# The Productivity of Cassava (*Manihot esculenta* Crantz) in Kagoshima, Japan, Which Belongs to the Temperate Zone

**Shin Yabuta [1], Tamami Fukuta [1], Shotaro Tamaru [2], Keita Goto [2], Yoshihiro Nakao [1], Phanthasin Khanthavong [2,3], Peter Ssenyonga [4,5] and Jun-Ichi Sakagami [1,\*]**

[1] Faculty of Agriculture, Kagoshima University, Kagoshima 890-0065, Japan; syabuta@agri.kagoshima-u.ac.jp (S.Y.); oritarin29@gmail.com (T.F.); nakao.yoshi.kagoshima@gmail.com (Y.N.)

[2] The United Graduate School of Agricultural Sciences, Kagoshima University, Kagoshima 890-0065, Japan; k5697087@kadai.jp (S.T.); k6988627@kadai.jp (K.G.); khanthavongp@gmail.com (P.K.)

[3] National Agriculture and Forestry Research Institute, Vientiane 7170, Laos

[4] Graduate School of Agriculture, Forestry and Fisheries, Kagoshima University, Kagoshima 890-8580, Japan; sspeter2@gmail.com

[5] National Agricultural Research Organization, Rwebitaba Zonal Agricultural Research and Development Institute, Entebbe P.O. Box 295, Uganda

\* Correspondence: sakagami@agri.kagoshima-u.ac.jp; Tel.: +81-99-285-8617

**Abstract:** The cultivation period of cassava in Kagoshima, Japan, which belongs to the temperate zone, is limited by the low temperature in winter. To maximize productivity under this limited period, investigations were conducted on the gas exchange rate and production structure relating to light utilization in a plant community of cassava grown under different nitrogen fertilization conditions. Fertilization either at planting or three months after planting significantly increased stomatal conductance in the upper canopy and root dry weight compared to the control. In addition, the dry matter distribution to stem and root dry matter rate of initial fertilization treatment were significantly higher, and the dry matter distribution to root of the latter fertilization treatment tended to be higher than that of the control. However, light transmittance at 80 cm below the top of the canopy was almost the same as that at the ground surface, which was a common tendency among the treatments. In conclusion, it was revealed that the effects of fertilization on yield were mainly the increase in the gas exchange rate of individual leaves and the change of dry matter distribution rather than an improvement in light transmittance.

**Keywords:** canopy structure; gas exchange rate; light transmittance

## 1. Introduction

Cassava (*Manihot esculenta*), a native crop to tropical America, is called a 21st century crop [1] because it has the ability to adapt to poor environmental conditions, such as semi-arid land, acidic soil, and low soil fertility. It has been widely used for staple foods, livestock feeds, processed foods, and starch production, mainly in tropical Asia and Africa [1]. Cassava tubers have a higher starch content than other tuber crops [2], and it was reported that the conversion efficiency of solar energy per unit land area to carbohydrates under favorable conditions exceeds that of maize and rice [3,4]. The global cassava production increased by 3.9 times between 1961 and 2018. By region, it increased by 5.4 times in Africa and 4.5 times in Asia. Compared to other major crops, cassava has the second highest growth rate after corn (5.6 times the growth rate during the same period). The harvest area of cassava expanded 2.6 times worldwide during the same period, with the largest increase rate in Africa (3.4 times), which has expanded rapidly since 2011, while the increase rate in other regions was relatively small (1.2 to 1.7 times). On the other hand, its yield increased by 1.5 times worldwide during the same period. By region, it increased by 2.7 times in Asia, followed by 1.6 times in Africa. From these trends in cassava production over the past half

century, it can be seen that production has increased due to area expansion in Africa and increased yields in Asia. Especially in Asia, it is evident that the yield increased sharply after 1995 [5]. The increase in cassava production has been supported by the development and dissemination of regional adaptable varieties centered on Centro Internacional de Agricultura Tropical (CIAT), the International Institute of Tropical Agriculture (IITA), and agricultural research institutes in each country. With the expansion of cassava production in recent years, cultivation cases have been reported in marginal regions utilizing the poor environmental resistance of cassava. Some examples have been reported, such as an investigation of the productivity of initial-maturing varieties using the fallow period after paddy rice cultivation in the tropical region of Asia [6], a productivity comparison in the high-altitude areas of tropical and subtropical regions [7,8], and the development of a resistant variety to low temperature stress that grows in the subtropical inland region during winter [9]. On the other hand, among the photosynthesis-related traits that form the basis of substance production, the maximum rate to receive the light energy projected on the crop canopy is as low as about 64% compared to soybean (90%), and the plant shape has been improved through breeding. The conversion rate from photosynthetic active radiation (PAR) to the biomass of cassava is around 1.4%, lower than that of soybean, which is 2.9~4.3%. Furthermore, El-Sharkwy and De Tafur [10,11] pointed out that no improvement in photosynthesis rate was observed between native and cultivars, and these results suggest that the previous breeding traits did not include photosynthesis-related traits.

Cassava cultivation in Japan is carried out only by small-holder farmers in the Kagoshima Prefecture island area and Okinawa Prefecture located in the southern part of the country. There are not many studies from the viewpoint of agronomics other than Minami et al. [12] and Minoda et al. [13]. Minami et al. cultivated cassava introduced from Brazil for three years in Kagoshima, which has an eight-month frost-free period from April to November, and reported that 20 t ha$^{-1}$ of raw potato yield, comparable to the yield of sweet potatoes cultivated in the same region, could be obtained. In Kagoshima Prefecture, sweet potato production is the highest in Japan and it is supplied to the starch, alcohol, and brewing industries. However, in recent years, the invasion of new pests and diseases that are thought to be related to rising temperatures has become a problem. [10]. As such, it has become important to search for new agricultural resources for regional industrial promotion as part of the climate change mitigation measures. At present, although the cultivation techniques and knowledge of cassava cultivation in Kagoshima are scarce, the fact that a certain yield has been obtained is expected to contribute to the expansion of cassava cultivation and production areas. On the other hand, there are many differences between the results of experiments conducted in the tropics and the results of Minami et al. [12]. For example, the leaf area index (LAI) was generally reported to reach its maximum at 4–5 MAP (months after planting) [14,15], but according to Minami et al. [12], the maximum LAI was measured 6 MAP. There was also big gap between their report, in which the three years average of harvest index was 0.3, and another report, in which it ranged between 0.46 and 0.71 including varietal difference [12,16]. Minami et al. [12] cite the effects of low sunshine and low temperature on this growth delay. The frost-free period in Kagoshima is about eight months, but it includes the period below the low temperature limit of cassava growth, which is said to be an average daily temperature of 18 to 20 °C [7]. In addition, the effects of heavy rainfall that occurs about 2 to 3 MAP and the accompanying low sunshine are also factors to consider. Therefore, it is necessary to clarify the growth characteristics of cassava in Kagoshima, which is a subtropical island region. In this study, among the photosynthesis-related traits that are the basis of the dry matter production of cassava, we focused on the canopy formation, the gas exchange rate of individual leaves, and the distribution of photosynthesis products. Therefore, this experiment was conducted to clarify the effect of the change in the community structure under different nitrogen fertilization conditions on the gas exchange rate, productive structure, dry matter production, and dry matter distribution (DMD). It was assumed that fertilization at the planting stage increases the gas exchange rate and promotes the developments of leaves and stems,

resulting in the accelerated establishment of a plant community. The increase of the total dry weight and root DMD is expected to contribute the increase of the root dry weight. On the other hand, we hypothesize that although fertilization at a later growth stage increases the photosynthesis rate, root DMD does not change since the dry matter will be distributed to the leaves, stems, and roots. Therefore, it is thought that only an increase of the total dry weight contributes an increase in the tuberous root dry weight.

## 2. Materials and Methods

### 2.1. Experimental Design and Sample Cultivation Site

The experiment was conducted at experimental field of field research center, faculty of Agriculture, Kagoshima University, Japan, in 2020. Nitrogen application (100 kgN ha$^{-1}$) was separated into three treatments with two different timings of application or no application. The initial and latter application treatments were on 22 May (0 DAT: days after transplanting) and 25 August (92 DAT), respectively. The application of 100 kg P$_2$O$_5$, K$_2$O ha$^{-1}$ was conducted on 22 May to all treatments including the control. The plots of 30 m$^2$ were stationed according to a randomized complete block design in three replicates. The cuttings were selected for uniform shoot and root development and transplanted vertically to soil ridges with a spacing of $0.5 \times 1$ m on 22 May. This spacing is common to avoid lodging by wind in Tokunoshima 600 km south of Kagoshima city where cassava cultivation occurs at a commercial scale. Weed control was conducted manually 1 and 2 MAP, until the plant canopy was established. Disease and insect pest were scouted at two-week intervals.

### 2.2. Preparation of Cassava Cuttings

One cassava genotype (Tokunoshima yellow) that is the most common variety in Tokunoshima was used in this experiment. It is necessary to explain stem preservation method in advance, because the cassava plant cannot survive outdoors during winter in Kagoshima. Branches harvested in December 2019 were pruned to 1.5 m length each and 20 tied stems were lowered into plastic containers filled up by river sand. They were covered by clear plastic sheets to maintain the temperature and humidity and kept in a plastic green house. Germinated and rooted parts were removed and stems with no roots and leaves were cut into 20 cm length and prepared for this experiment. First, 20 cuttings were planted to plastic pot (20 cmW $\times$ 60 cmL $\times$ 20 cmD) filled with vermiculite on 4 April 2020, and then they were transplanted into the experimental field on 22 May 2020.

### 2.3. Gas Exchange Measurement

Gas exchange parameters were evaluated using a portable gas exchange measurement system (LI-6400, Li-Cor Inc., Lincoln, NE, USA) equipped with the standard leaf chamber (chamber area of 6 cm$^2$) and porometer (AP-4, Delta-T Devices, Cambridge, UK). A layer height of 40 cm was established from the top of the ridge to the top of the canopy in the center of each plot. All gas exchange measurement of the fully expanded leaves were conducted at each layer on sunny days (8:00–13:00). First, the photosynthetic rate (*A*) and transpiration rate (*E*) were measured on 7 August to investigate maximal values for leaves at different heights in the canopy. The measurement settings included a light intensity of 1200 µmol m$^{-2}$ s$^{-1}$, an ambient CO$_2$ concentration of 420 µmol mol$^{-1}$, and a block temperature of 32 °C. A second measurement was conducted on 14 August to test the photosynthetic response to the light intensity. Then, the chamber irradiation intensity was changed into different 11 levels between 0 to 1200 µmol m$^{-2}$ s$^{-1}$. The plant height reached more than 160 cm after September and the photosynthesis measurement was stopped. The measurement of stomatal conductance (*gs*) continued until end of the experiment in November. The *gs* measurement started on 24 June and continued with a one-month interval in September when the damage from a typhoon persisted. Plant height was measured from the top of the canopy to the top of the ridge to determine the layers for gas exchange measurements at different heights in the canopy.

*2.4. Measurement of Canopy Structure*

The measurements of PAR were conducted using the quantum sensor (MIJ-14PAR, Environmental Measurement Japan, JAPAN). A set of data was recorded by two quantum sensors fixed on steel pole with 20 cm distance at the top of each layer on and between the ridge per plot. Measurements were conducted on 25 August at the middle growth stage and on 16 November before the canopy structure was evaluated by the stratified clip method. Cloudy days were selected to avoid the influence of direct skylight, and 1.5 $m^{-2}$ sampling areas were set per plot for all treatments and the stratified clip method was carried out. Plants inside the sampling area were divided into an underground part and six layers of 40 cm height parallel with the ground surface. The plant body contained in each layer was divided into leaf, petiole, and stem, and the underground part was separated into fine root, tuberous root, and stem. All plant parts were measured in terms of their flesh weight, and then a portion of them were taken for drying and we measured flesh weight again, because even individual plant parts were too large for our oven. All sample parts were dried in oven set at 80 °C for three days and then the leaf samples were measured for their area using an automatic area meter (AAM-9, Hayashi Denko Co., Ltd., Osaka, Japan). Afterwards, their dry weight was measured. The dry matter rate (DMR) of each part and specific leaf area (SLA) per layer were calculated by the following Formulas (1) and (2):

$$\text{DMR (\%) = organ flesh weight in layer/organ dry weight in layer} \times 100, \tag{1}$$

$$\text{SLA (cm}^2\,\text{g}^{-1}\text{) = leaf sample area/leaf sample dry weight,} \tag{2}$$

The dry weight of individual organs was calculated for each layer as the organ flesh weight multiplied by the DMR of the target organ. Leaf area was also calculated for each layer as the leaf dry weight multiplied by SLA at the target layer. The dry matter weight for each layer was calculated by multiplying the fresh weight of the organ for each layer by its DMR. Finally, the total dry weight was expressed as the sum of the dry weights of the layers. Three plants were taken from two plots of each treatment, then they were separated into the parts mentioned above, and flesh weight was measured. The total dry weight was calculated as the sum of the dry weights of individual organs resulting from multiplying the DMR by the flesh weight of each organ.

*2.5. Weather Data*

The average, minimum, and maximum air temperature, daily total solar radiation, and precipitation during the experimental period were recorded 2 km south of the experimental site, which is managed by the Japan Meteorological Agency. Data were obtained from the Japan Meteorological Agency website.

*2.6. Data and Statistical Analysis*

Comparisons between the control and initial fertilization treatments for mean *A*, *gs*, intercellular $CO_2$ concentration (*Ci*), *E,* and light transmittance rate on August were done by using paired *t*-test. Mean comparisons of plant height, *gs* at different canopy levels, LAI, dry weight and DMD were done using one-way ANOVA for comparing the means among treatments. Averages of multiple comparisons were determined by Tukey's test under BellCurve for Excel (Social Survey Research Information Co., Ltd., Tokyo, Japan). The correlation analysis between *A* and *Ci* were analyzed. Statistical significance was taken at $p < 0.05$ for all analyses.

**3. Results**

*3.1. Weather Conditions at the Experimental Field*

For the cassava planted in May 2020, the daily average, maximum and minimum temperatures continued to increase from beginning of May to August and they reached 31.4, 37.0 and 28.5 °C, respectively (Figure 1). After that, they began to decrease and were below 15, 20, and 10 °C in end of November, respectively. The total precipitation

during this experiment was 2453.5 mm, and in June and July, the rainfall was over 700 mm. On the other hand, there were only two rainy days from 28 July to 20 August and the rainfall during this period was 42 mm. Daily solar radiation was greater than 25 MJ m$^{-2}$ on sunny days, but the monthly solar radiation ranged 116.5–177.5 MJ m$^{-2}$ from May to July when cloudy or rainy days occupied the majority of this period. However, there was high monthly solar radiation in August (265.4 MJ m$^{-2}$) and October (219.4 MJ m$^{-2}$) with low monthly precipitation. The meteorological conditions in Kagoshima from June to December 2020, when the experiment was conducted, were compared with the average year data. Although the temperature and total solar radiation fluctuated slightly from month to month, the total values during experimental period were 100.6 and 98.5% of the average year, almost the same levels. However, the rainfall pattern in Kagoshima during this season is greatly affected by the intensity of the rainy season and potential typhoons. The total precipitation in this period was 31.7% more than the average year since heavy rain persisted from the end of June to the beginning of July. Although there was a typhoon on September, weather condition from September to November were stable and total precipitation, temperature, and solar radiation ranged between 99.4 and 103.6% of the average year, respectively.

### 3.2. Gas Exchange Parameter

Photosynthetic measurements in the control and initial fertilization treatment were performed at three and four layers, respectively. The plant height at the measurement time varied between treatments (Figure 2). The maximum photosynthetic rates ($A_{max}$) of the control and initial fertilization were 21.3 and 25.1 µmol m$^{-2}$ s$^{-1}$ under 1200 µmol m$^2$ s$^{-1}$ of PAR at the highest canopy layer, respectively. As the measured layers were lower, their $A_{max}$ and light saturation point became low. Both $E$ and $gs$ showed a similar tendency of having higher values at higher layers, but they did not show saturation. The intercellular $CO_2$ concentration ($Ci$) under the dark ranged from 387 to 421 µmol mol$^{-1}$ and as light intensity increased, it decreased to around 290 µmol mol$^{-1}$. Photosynthetic parameters were measured with 1200 µmol m$^{-2}$ s$^{-1}$ of PAR to compare between treatments (Figure 3). Basically, photosynthetic parameters at 40–120 cm height levels showed that the control had higher values than the initial fertilization treatment. However, as a result of the comparison at the highest canopy layer of both treatments, $A$, $gs$ and $E$ at the initial fertilization treatment were greater than the control. It is known that the slope of regression line between $A$ and $gs$ indicates intrinsic water use efficiency (iWUE) [17]. There were significant liner relationships between the $gs$ and $A$ of both treatments, and the slope at the initial fertilization treatment was bigger than at the control (Figure 4).

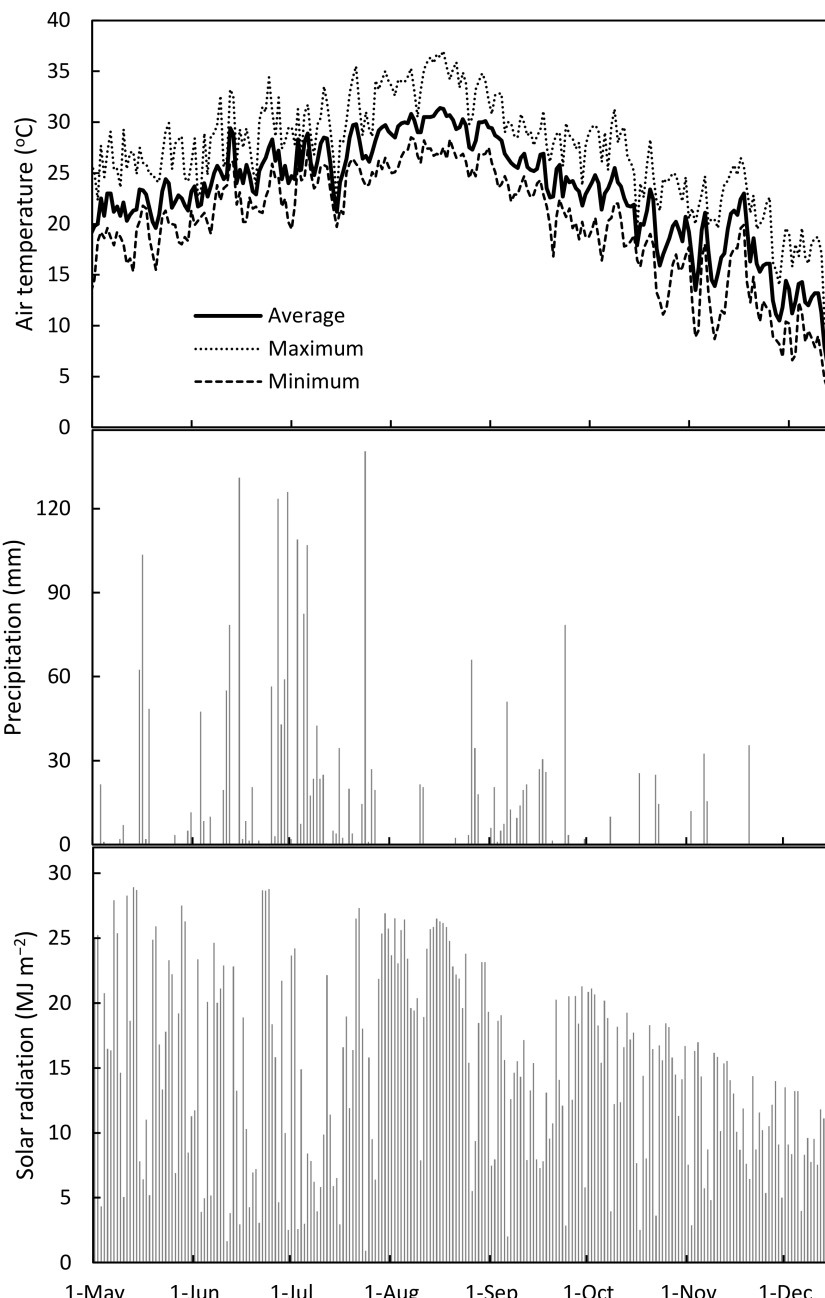

**Figure 1.** Minimum, maximum, and average air temperatures, daily precipitation, and solar radiation in the experiment period.

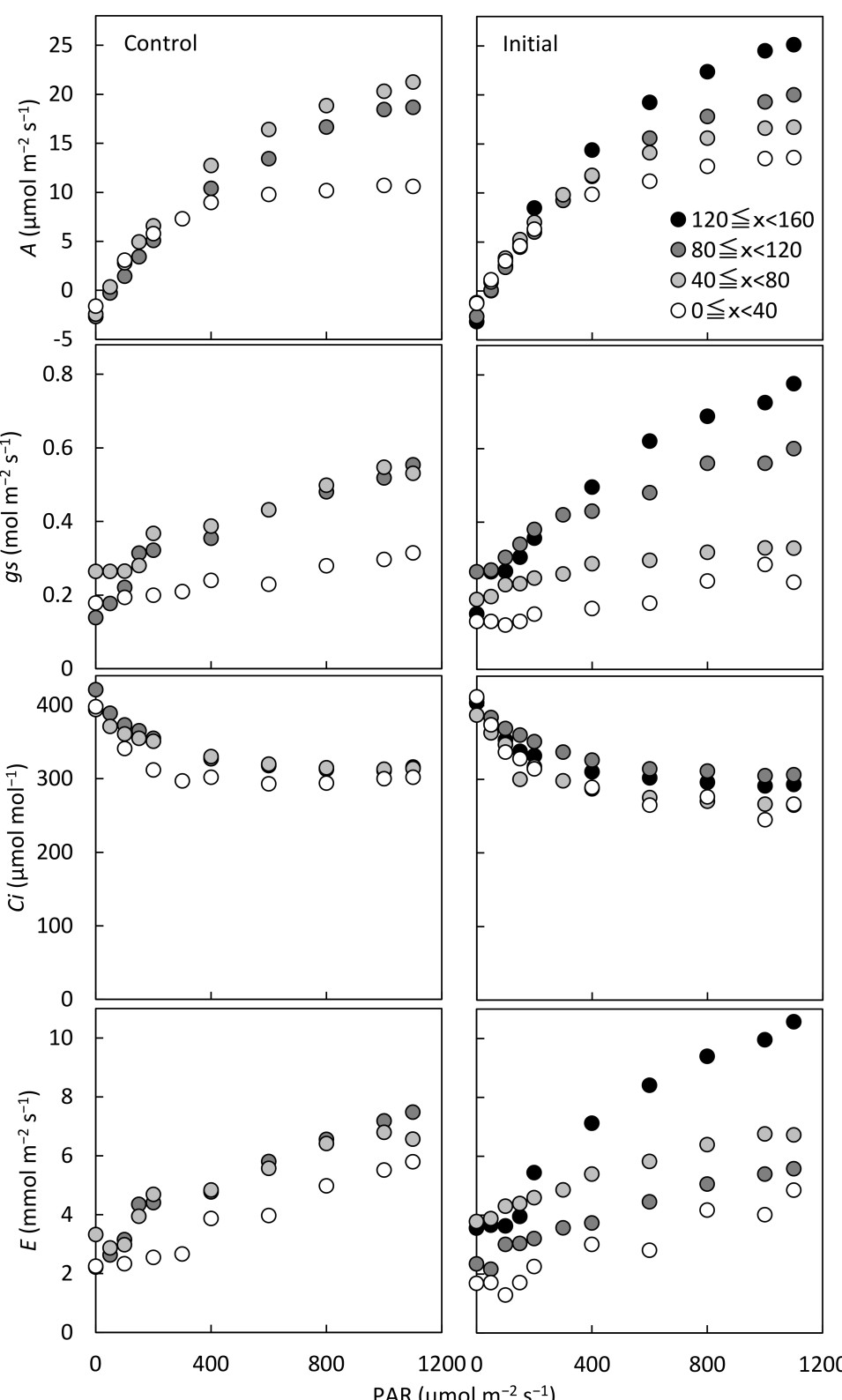

**Figure 2.** Responses of photosynthetic rate (*A*), stomatal conductance (*gs*), intercellular $CO_2$ concentration (*Ci*), and transpiration rate (*E*) to photosynthetic active radiation (PAR) at different levels of the crop community.

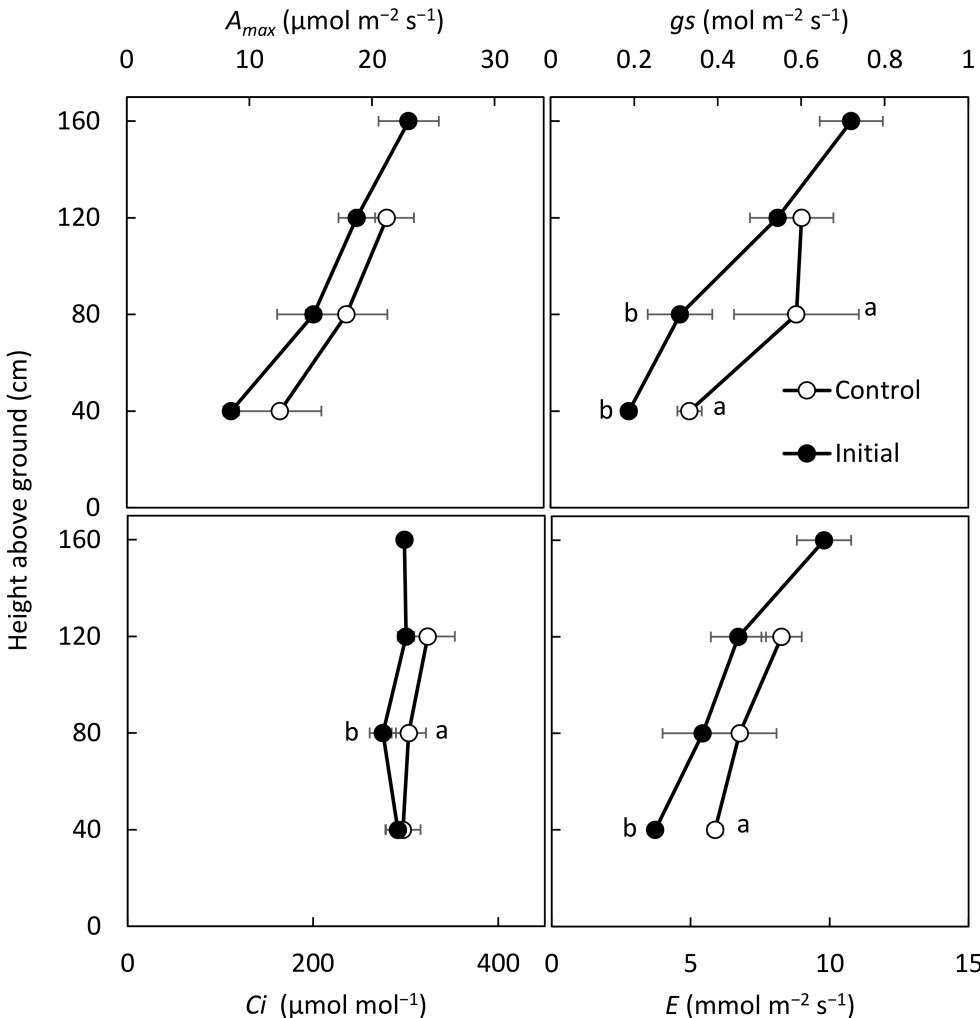

**Figure 3.** Effect of the control (○) and initial fertilization treatment (●) on maximum photosyn-thetic rate ($A_{max}$), stomatal conductance ($gs$), intercellular CO2 concentration ($Ci$), and transpiration rate ($E$) at different heights of the plant community. Different letters indicate statistical significance at $p < 0.05$.

### 3.3. Stomatal Conductance

Plant height was measured to determine the number of layers for $gs$ measurements, and it continued to increase during the vegetation period (Figure 5). Plant height in the initial fertilization treatment was significantly higher than in the control after August. A significant difference of plant height between the latter fertilization treatment and the control was observed only in November. As the result of gs measurements in June, both treatments showed high values from 656.0 to 730.0 mmol m$^{-2}$s$^{-1}$ and there were no significant differences between them for all the layers (Figure 6). Plant height topped 1.2 m and there were three layers in the canopy in July. In the upper two layers, $gs$ was significantly differentiated between treatments, and the initial fertilization treatment was greater than that of the control. However, there was no significant difference on the lower layer due to light shortage, as self-shading had begun. There were three layers in August, but leaf distribution ranged in height. Leaves were located from 80 to 200 cm in the initial fertilization treatment and from 40 to 160 cm in the control. There were no leaves under these layers due to defoliation. There were significant differences between treatments only in the layers 80–120 cm above the ground. In October, plant height top exceeded 200 cm in all treatments, and there were significant differences among treatments in gs at the top layer. In November, new leaves emerged from the vestige of defoliation in layers below

80 cm above ground, but the gs of most of these leaves was less than 100 mmol m$^{-2}$ s$^{-1}$. The gs of the latter fertilization treatment was significantly higher than the control in layers above 160 cm.

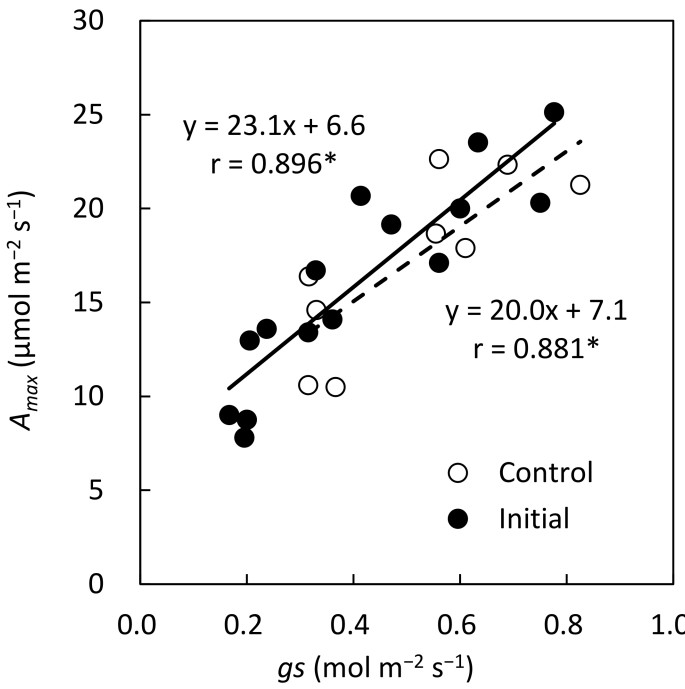

**Figure 4.** Relationship between maximum photosynthesis rate ($A_{max}$) and stomatal conductance ($gs$) on the control (○) and initial fertilization treatment (●). Significant correlation ($p < 0.05$) is indicated with *.

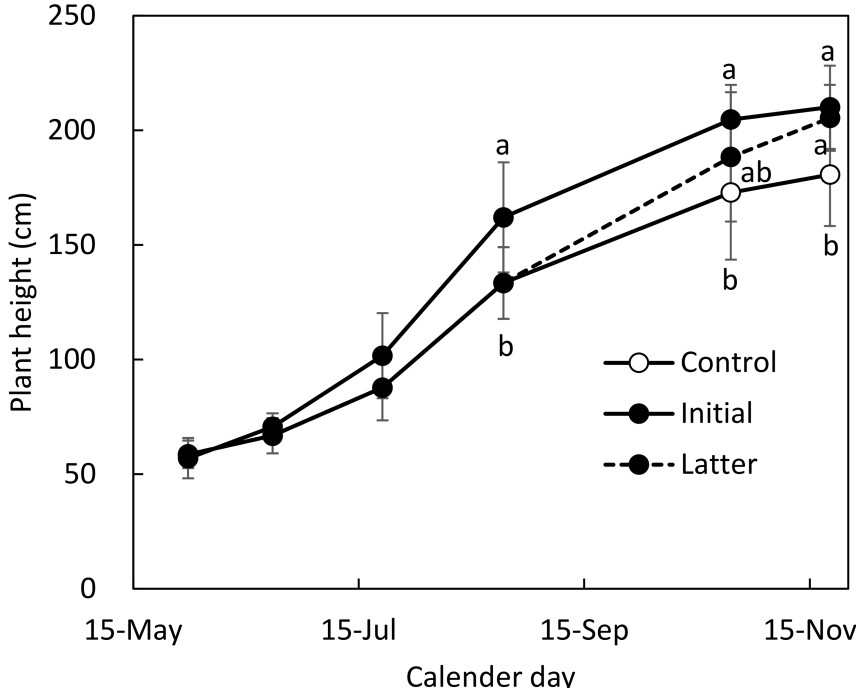

**Figure 5.** Changes in plant height. The value shown is an average of readings from the control (—○—), initial (—●—), and latter fertilization treatments (–●–). Different letters represent statistically significant differences ($p < 0.05$) among the treatments.

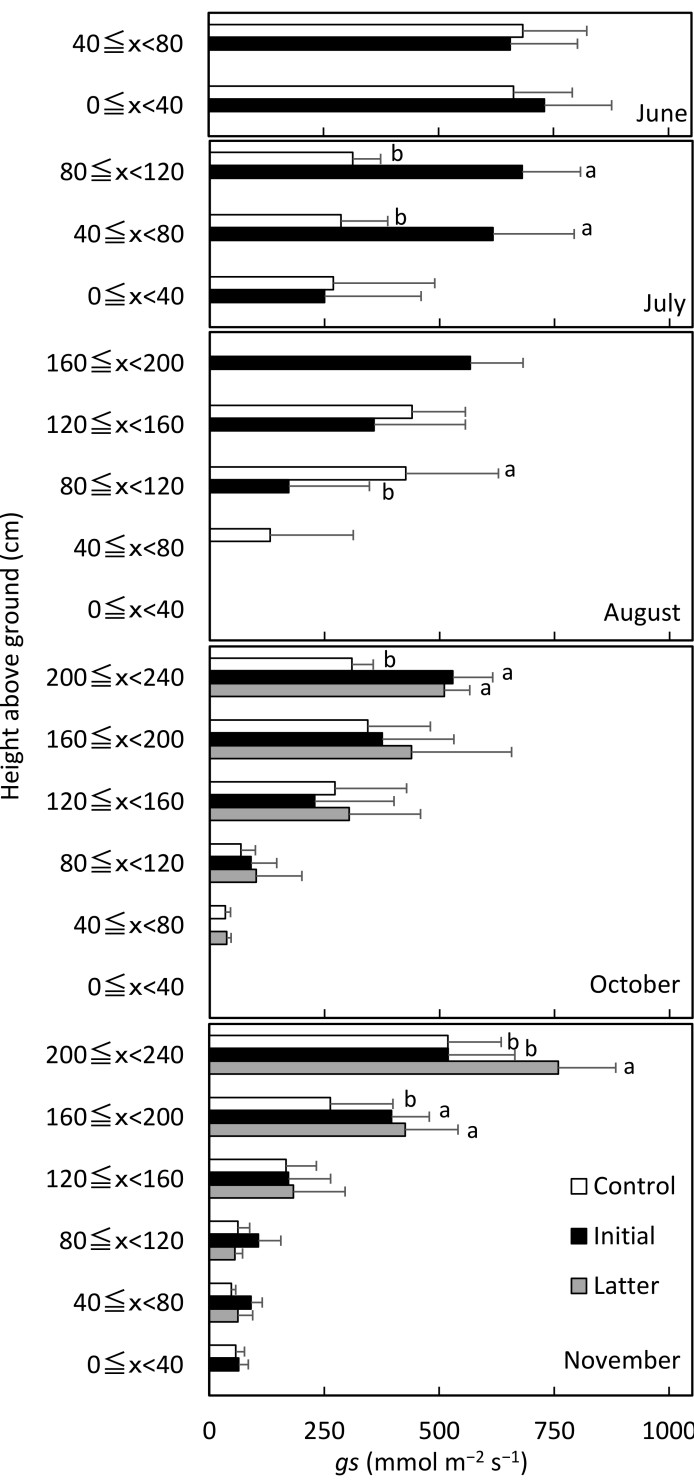

**Figure 6.** Stomatal conductance (*gs*) of leaves at different canopy levels with 40 cm interval from June to November. The value shown is an average of readings from the control, initial, and latter fertilization treatments. Different letters represent statistically significant differences ($p < 0.05$) among the treatments.

### 3.4. Leaf Distribution and Light Transmittance Rate

In August, plant height in the control and the initial fertilization treatments was 133.3 and 162.0 cm, respectively. Four and five layers were set up in each treatment, and light transmittance in all the layers was measured (Figure 7). The height of the layer in which light transmittance started to decrease was different, but there was no difference

at the 80 cm height level (50.6 and 44.7%). Light transmittance decreased slightly and was 43.3 and 44.5% at the ground surface in the initial fertilization treatment and the control, respectively. There were five layers from 40 to 240 cm above ground at the initial fertilization treatment and six layers from ground surface to 240 cm at the control and latter fertilization treatments, respectively (Figure 8). The peak leaf area per layer on initial fertilization was located on the second layer from the top of the canopy, but the third layer of the control and latter fertilization were the largest. The LAI of each treatment was as follows: 3.91, 4.19, and 5.44, with the highest and lowest values seen at the latter and initial fertilization treatments, respectively (Table 1). Light transmittance rates on the ground surface were 27.5% (initial), 28.3% (latter), and 33.6% (no), but there was no change in light transmittance even at a 120 cm height level. The decrease of light transmittance occurred at a height between 120 and 240 cm of the canopy. Therefore, as the result, the investigation focused on the three layers from the top of the canopy. There were differences among the treatments in leaf area distribution and in the decline of the light transmittance rate on each layer. The leaf area in the top layer was only 517 cm$^2$ in the control and the decline of light transmittance to 93.9% was the smallest among the treatments. On the other hand, the leaf area at the second highest layer was 10680 cm$^2$ and light transmittance rate rapidly dropped to 38.6%. The leaf area in top of layer in initial fertilization treatments was 4078 cm$^2$, which was the largest among the treatment and the decline in light transmittance was also the biggest, dropping to 50.0%. The leaf area in second layer was 19013 cm$^2$, and the light transmittance decreased to 33.3%. There were 2899 and 18569 cm$^2$ leaf area on first and second layer of latter fertilization treatment, respectively, and both were the second biggest among the treatments. The light transmittance rate decreased to 61.2 in the first layer and 36.6% in the second layer.

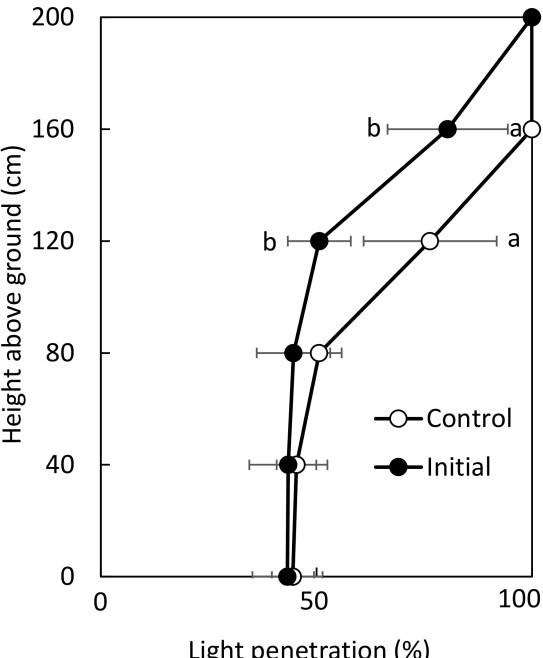

**Figure 7.** Light transmittance rate through different canopy levels in August. The value shown is an average of readings from the control (○) and initial fertilization treatment (●). Different letters represent statistically significant differences (*p* < 0.05) between the treatments.

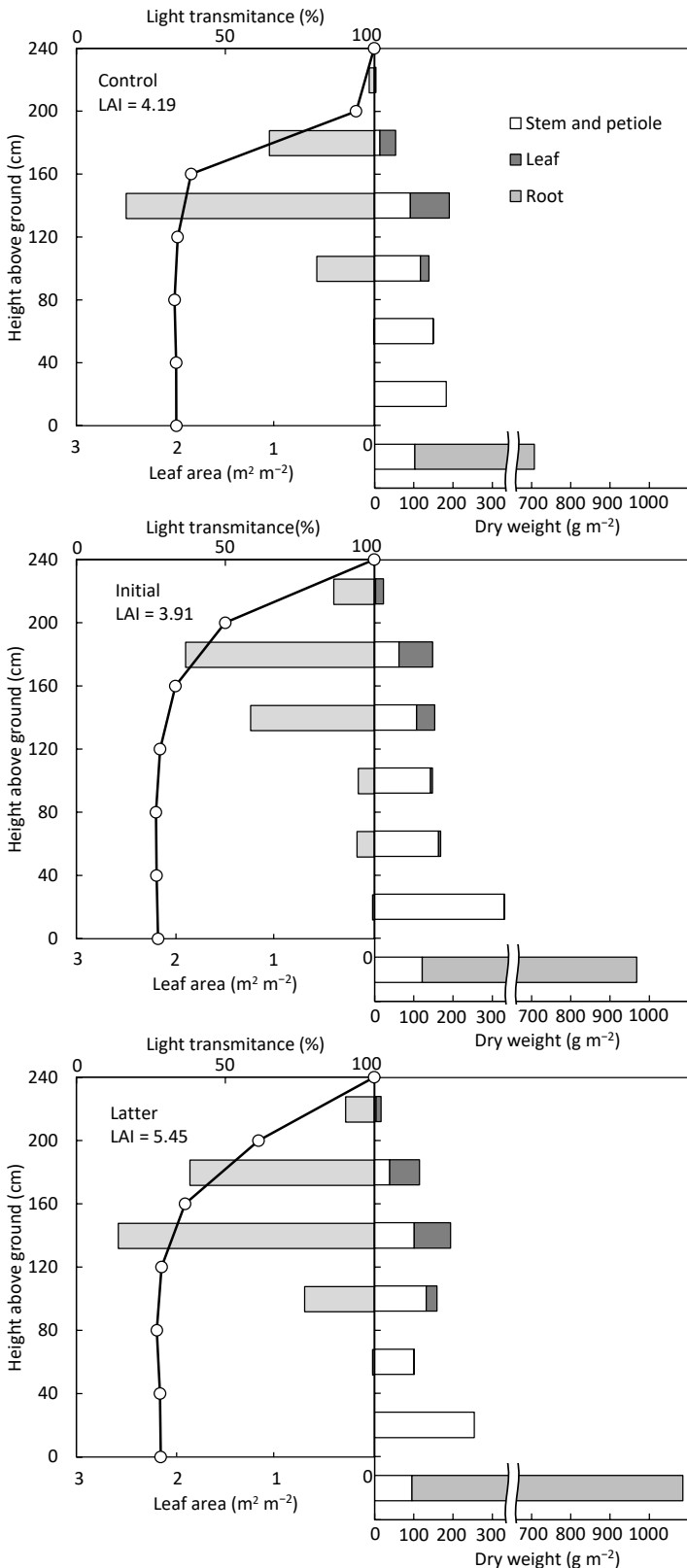

**Figure 8.** Light transmittance rate (○), leaf area, and dry matter distribution of the control, initial, and latter fertilization treatments.

**Table 1.** LAI, dry weight, and dry matter distribution of plant parts and tuberous root parameter including flesh yield and root dry matter rate with different fertilization conditions.

| Treatment | LAI | | Leaf | | Petiole | Stem | | Root | | Total | | Tuberous Root | | |
|---|---|---|---|---|---|---|---|---|---|---|---|---|---|---|
| | | | | | Dry Weight (gm$^{-2}$) | | | | | | | Yield (gFWm$^{-2}$) | Dry Matter Rate (%) | |
| Control | 4.19 | b [2] | 161.2 (10.6) | b | 55.2 (3.6) [1] | 601.4 (39.5) | b AB | 704.4 (46.3) | b | 1522.2 | b | 2535.2 | b | 27.4 | ab |
| Initial | 3.91 | b | 162.8 (8.0) | b | 59.1 (2.9) | 871.0 (42.7) | a A | 946.5 (46.4) | a | 2039.4 | a | 2888.6 | b | 32.5 | a |
| Latter | 5.45 | a | 209.2 (10.3) | a | 66.3 (3.3) | 659.8 (32.6) | b B | 1089.7 (53.8) | a | 2024.9 | a | 4099.6 | a | 26.5 | b |

[1]: Values in brackets are DMD of plant organs. [2]: Different letters represent statistically significant differences ($p < 0.05$) among the treatments.

### 3.5. Dry Matter Production and Yield Parameter

The leaf dry weight in the latter fertilization treatment (209.2 g m$^{-2}$) was significantly larger than in the other treatments (161.2, 162.8 g m$^{-2}$), but the DMD of leaves ranged between 8.0 and 10.6% and there was no significant difference among the treatments (Table 1). The stem dry weight at initial fertilization (871.0 g m$^{-2}$) was the biggest among the treatments, and DMD (42.7%) was also bigger than it was at the latter fertilization treatment (32.6%). The root dry weight at the initial fertilization treatment (946.5 g m$^{-2}$) tended to be greater than at the control (704.4 g m$^{-2}$), but the rate of the root dry weight per total dry weight was 46.4 and 46.3%, respectively, almost the same. On the other hand, the root dry weight at the latter fertilization was 1089.7 g m$^{-2}$ and it made up 53.8% of the total dry weight. The total dry weight of the control (1522.2 g m$^{-2}$) was significantly smaller than it was at the initial and latter fertilization (2039.4 and 2024.9 g m$^{-2}$). The flesh yield of the tuberous root at latter fertilization was significantly largest among the treatments. However, the DMR of the tuberous root at the latter fertilization (26.5%) was smaller than at the initial fertilization (32.5%).

### 4. Discussion

First, the effects of the weather conditions in 2020 on the result of this study were examined. The total temperature and total solar radiation were at the same levels as the average year, but the total precipitation was higher than that at the average year due to the effect of the rainy season in June and July. However, it is assumed that the rhizosphere of cassava was not submerged because the cuttings were transplanted on top of the ridges. The total solar radiation in July was 76.3% of the average year, because of the low insolation during the rainy season. Although there was significantly higher gs at the initial fertilization treatment on July, it is probable that the effect of fertilization on gs was limited, since the stomata does not open under low solar radiation. However, the total dry weight of the initial fertilization treatment was significantly higher than that of the control, so it is considered that the effects of the low insolation and high precipitation in July on the conclusions of this study are not strong. Since stable climatic conditions continued after September, except for the typhoon, the effect of the latter fertilization treatment is expected to be the same as the average year conditions.

In this study, cassava of tropical origin was cultivated in Kagoshima, which belongs to the temperate zone, under different fertilization conditions, and its growth characteristics were investigated from the viewpoint of individual leaf gas exchange rate, plant community structure, and dry matter production and distribution. The initial fertilization promoted an increase in plant height from the initial growth stage, resulting in early canopy closure. The increase of plant height on latter fertilization continued in October and November, when the plant height increases of the control and initial fertilization declined. These results indicated that fertilization at different growth stages can affect canopy development.

The response of leaves in the upper layer, which has high photosynthetic activity, to light intensity was compared with previous reports. While the $A_{max}$ in this study was 21.3–25.1 μmol m$^{-2}$ s$^{-1}$, Santanoo et al. [18] and Mahakosee et al. [19] reported it to be

approximately 30 $\mu$mol m$^{-2}$ s$^{-1}$. PAR at a light saturation of $A$ was also reported to be about 1500 $\mu$mol m$^{-2}$ s$^{-1}$, higher than the 1200 $\mu$mol m$^{-2}$ s$^{-1}$ reported in this study. One of the reasons for this is that Kagoshima, where this experiment was conducted, is at a higher latitude and receives less solar radiation than Khon Kaen, Thailand, where Santanoo et al. [18] conducted their experiments. The result of light curve on different canopy height levels showed high $A_{max}$ on the upper layer and photosynthesis was conducted under high solar radiation. On the other hand, although the lower leaves had a low $A_{max}$, they showed the same level of $A$ as the upper leaves, when the PAR was less than 400 $\mu$mol m$^{-2}$ s$^{-1}$. This demonstrates that they adapted to the low solar radiation environment inside the crop community. This result is also consistent with the report by Santanoo et al. [18], which divided the canopy into six layers and measured $A$ in four varieties of cassava. A comparison of the photosynthetic parameters during the light saturated condition in August (Figure 3) showed that there were no significant differences at each height level between treatments. This was the result of adaptation to the degraded light environment, as the initial fertilization promoted increased plant height and new leaf expansion, causing self-shading. In addition, there were significant correlations between gs and $A$ at initial fertilization and the control, and it was suggested that the $A$ can be estimated using gs as an index even under different fertilization conditions (Figure 4). The effect of initial fertilization on gs was apparent in July and August, especially in the top canopy layer exposed to high solar radiation. There were significant differences in gs in October and November between the latter fertilization treatment and the other treatments. These results make it clear that fertilization in the initial and latter growth stage of cassava improves photosynthetic activity on the upper layer of the crop community.

According to the measurements of light transmittance in the crop community in August, the height of the top layer was different between the control and initial fertilization treatments (Figure 7); light transmittance dropped sharply to 40% below the second layer and did not change between the third layer and ground surface. However, there were differences in the layer heights and light transmittance decreased rapidly among all treatments in November. A sharp decrease in light transmittance of about 50% was observed in the second layer of the control and in the first layer of the initial fertilization plots. It was shown that only the leaves in the upper part of the canopy had sufficient sunlight. On the other hand, in the latter fertilization plots, light transmittance decreased by 38.8% and 24.6% at the first and second layers, respectively, indicating that the light reached the inside of the canopy. However, the maximum leaf area per layer was distributed in the third layer in the control and latter fertilization plots and in the second layer in the initial fertilization plot, below the layer where the transmittance was sharply reduced. Several research works have reported that LAI maximizes the yield by 2.5 to 3.5 times [20–22]. The LAI of this study was between 3.91 (initial fertilization) and 5.45 (latter fertilization). This suggested that excess leaves may have grown in the community. In this study, the decrease in light transmission occurred in the first and second layers, which was common in both the August and November measurements. However, according to Santanoo et al. [18], the decrease in light transmittance occurred in lower layers at 6 MAP than at 3 MAP, and the minimum light transmittance was smaller at 6 MAP, suggesting that the community was still growing after 3 MAP. This may be because the planting density of Santanoo et al. [18] was lower than that of this study, and thus it may have taken longer to close the community.

A comparison of total dry matter weights revealed that both initial and latter fertilization had a positive effect on dry matter production and that the application time influenced DMD. The root DMD of the control and initial fertilization treatment were almost the same, 46.3 and 46.4%, respectively, and that of latter fertilization treatment was slightly higher at 53.8% (Table 1). It is known that root DMD, an important factor related to yield, is greatly influenced by the cultivar, cultivation period, and environment. There are reports of root DMD of 30.1% to 31.6% [23] and 49% to 56% [24] for cultivation periods of around 180 days, so the root DMD in this study is not lower than that of other reports for similar cultivation periods. Minami et al. [12] also conducted cassava cultivation at Kagoshima with almost the

same cultivation period as this research for three years. However, the root DMD was lower than it was in this study, about 30%. Lian and Cock [25] stated that although the roots and the above-ground part grow at the same time, the dry matter is preferentially distributed on the above-ground part until canopy development finishes. It is purported that the period of preferential distribution to the roots was short, and the harvest index remained low in Kagoshima, where the growing period is limited. The plant density of Minami et al. [12] was 1.23 to 1.30 hill m$^{-2}$, lower than in this study, which imitated Tokunoshima's plant density, which is a cassava producing area in Kagoshima. As a result, we believe that the root DMD became low because it took a long time for the development of the plant community. Our first hypothesis was that the early enlargement of tuberous root leads high root DMD. However, the root DMD of initial fertilization treatment did not change but rather the stem DMD showed the higher tendency. These results suggest that although the initiation of tuberous formation was promoted by initial fertilization treatment, the dry matter accumulation in the stem before it was large and they were maintained until harvest. But in latter fertilization treatment, due to not only the increase of gs in October and November, but also the distribution of photosynthetic products focusing on root by fertilization in August, when the plant community was developed, a high DMD was observed. On the other hand, there is a report of 60–84% after ten months of planting [10], suggesting that further improvement is possible if the cultivation period at Kagoshima can be extended. The DMR of tuberous roots was significantly higher at the initial fertilization treatment (32.5%) than in the other two treatments (26.5, 27.4%). It is suggested that starch accumulation may have started earlier at the initial fertilization treatment where the plant height increased faster, and the canopy was closed earlier. Since the dry matter weight and DMD of the stems are significantly higher at the initial fertilization treatment, fertilization at the initial growth stage, when starch accumulation has not started, increases the dry matter production, but the effect of the increasing yield may be limited due to increasing the distribution to the stem. The effects of growing period, cultivar, management, and precipitation during the growing period on root DMR have been reported [26,27], but most of the reports discussed the results of samples around 12 MAP. Variety is one of the important factors affecting root DMR, which has been reported to be a minimum of 27.49–31.9% and a maximum of 41.19–41.7% [11,28,29]. The highest dry matter rate of root was 32.5% at initial fertilization in this study, and was low compared to other reports. Lian and Cock [25] stated that tuberous root enlargement occurs after the development of above-ground parts. In this study, since the canopy was not fully developed until August, the actual tuberous root enlargement period may have been even shorter. In particular, the control and latter fertilization treatments, which lacked fertilizer in the early growth stage, delayed community development and the initiation of tuber hypertrophy, resulting in a decrease of tuber DMR (Table 1). In addition, the latter fertilization resulted in significantly higher fresh yield than the other treatments due to the tuber enlargement caused by vigorous photosynthesis after fertilization, but the DMD of tuberous root was probably lower because there was not enough time to fill the enlarged sink capacity.

The summary of the mechanisms by which each fertilization treatment increased the root dry weight is as follows. Both the initial and latter fertilization increased total dry weight by promoting the gas exchange rate on the upper layer of the plant community, but there was no difference in the utilization of solar radiation associated with the change in productive structure in any treatments. The increase in root dry matter weight was achieved not only by increasing the gas exchange rate, but also by increasing root DMR due to promoting tuber formation in the initial fertilization. In case of the latter fertilization treatment, a high gas exchange rate and increasing DMD to the root contributed to the high root dry weight.

Although there are not many reports investigating cassava production in Kagoshima, the tuberous root dry weight of this study was 694 g–1087 g m$^{-2}$ (6.9–10.8 t ha$^{-1}$), larger than the data reported by Minami et al. [12], whose yield ranged 2.30~5.24 t ha$^{-1}$. An important difference in the cultivation management of these studies was the plant density,

which was almost twice as high in our study as in Minami et al.'s study [12]. This suggests that the high planting density may have caused early canopy closure and enlargement of the tuberous root. It is difficult to compare yields at different regions because cassava yield is easily affected by cultivation period [10,30], genetic background [31], and cultivation methods [27]. However, we selected studies with a cultivation period of about six months and compared them with the yield data of this study. Howeler and Cadavid [15] and Connor et al. [23] compared and evaluated the yield of two cassava cultivars, M Mex59 and M Col22, and their yield ranged 2.0–3.0 t ha$^{-1}$ and 4.0–6.2 t ha$^{-1}$, respectively. Hammer et al. [30], who tracked the yield with four different planting periods, reported a yield of 7.0–13.0 t ha$^{-1}$, if the low temperature period was not encountered immediately after planting. Lenis et al. [31] grew 1350 plants of cassava clone which had different two type of leaf retention ability and reported that the clone group with leaf retention ability (6.95–9.16 t ha$^{-1}$) had a significantly larger yield than the clone group without it (5.10–7.09 t ha$^{-1}$) in both March and May. As for the dry matter production in this study, it was clarified that the productivity of cassava within 6 MAP in Kagoshima is comparable to that in the tropical region. On the other hand, it is known that tuberous root dry matter continues to increase after six months and it increased by 2–3 times at 10–20 MAP [15,23,30]. Irikura et al. [7] reported that the temperature limit of cassava growth was 18–20 °C daily mean temperature and observed that cassava growth stopped at Kagoshima after mid-November, when the daily mean temperature dipped below 15 °C [12]. Thus, Minami et al. [12] concluded the period in which cassava can grow in Kagoshima is around eight months, from May to November. Although the period that cassava cannot grow is one of the limiting factors for the consideration to introduce various cultivation methods into Kagoshima, Singh et al. [32] conducted a cultivation experiment in Northern India, which has low temperatures in the winter and hot weather in the summer and reportedly obtained 7.5–12.3 t ha$^{-1}$ yield in terms of dry weight in the eight-month cultivation period. Although the cultivation results, techniques, and knowledge accumulated in Kagoshima are insufficient, the fact that a certain yield was obtained in that study suggests that there is much room for yield improvement by accumulating experience and introducing new techniques and knowledge into cassava production. As the cassava cultivation area expands, it is estimated that the use of the edge area in which environment is not favorable for cultivating cassava will increase. All regions have their own problems; therefore, the accumulation of knowledge and cultivation examples in these regions, including in this study, is expected to contribute to maximizing the yield in similar areas.

## 5. Conclusions

The initial and latter fertilization treatments significantly increased the total and root dry weight, but the parameters associated with them differed between treatments. In the initial fertilization, the increase in root dry matter weight was achieved not only by increasing the gas exchange rate, but also by increasing root DMR due to promoting the initiation of tuber formation. In the latter fertilization, the period when the gas exchange rate was high and the period when tuber was enlarged overlapped with each other. Therefore, when the DMD to root increased, the dry matter weight of root also increased. Community structure, photosynthesis of individual leaves, and DMD to root are known to be important for cassava tuberous root production. However, their evaluations are not always the same, so their importance may change depending on the cultivation environment.

**Author Contributions:** Conceptualization, S.Y. and J.-I.S.; methodology, S.Y. and T.F.; validation and analysis, S.Y. and P.K.; investigation, S.Y., S.T. and K.G.; data curation, S.Y.; writing—original draft preparation, S.Y.; writing—review and editing, S.Y., P.S. and Y.N.; supervision, J.-I.S.; project administration, S.Y. All authors have read and agreed to the published version of the manuscript.

**Funding:** This research received no external funding.

**Institutional Review Board Statement:** Not applicable.

**Informed Consent Statement:** Not applicable.

**Data Availability Statement:** No new data were created or analyzed in this study. Data sharing is not applicable to this article.

**Conflicts of Interest:** The authors declare no conflict of interest.

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
