# Peer review of "The Productivity of Cassava (Manihot esculenta Crantz) in Kagoshima, Japan, Which Belongs to the Temperate Zone"

_agronomy, doi:10.3390/agronomy11102021_

Round 1
Reviewer 1 Report
The purpose of the study was the clarifying N fertilization timing of the cassava var. Tokunoshima yellow cultivated in Kagoshima, Japan, based on its effects on gas exchange parameters, canopy structure and absorbance, and dry matter production and distribution. The experiment was conducted at experimental field of Kagoshima University in 2020 year. The experimental treatments included 3 N application times: initial (May 22th, at planting time), latter (August 25th) and no application (control). The experiment was set up according to randomized complete block design, with 30 m2 plots and 3 replicates.
The manuscript is written with the English language including many grammar mistakes, which make the text difficult to follow. Most of the reviewer comments and suggestions are included in the attached text. However, the Introduction sufficiently well describes the background of the study and uses relevant references. Material and Methods chapter needs to be corrected. I would suggest changing the name of the “no application treatment” into just “control treatment” or “control” and to precise N rate in the other treatments and explanation it’s the base (how it was calculated). PAR parameter should not be recognized as “morphological” one and moved to other sup chapter. Among many mistakes one is necessary to explain: the photosynthesis parameter measured with LI-6400 Photosynthesis System is named by the author as just “photosysnthesis (A)” or “Maximum net leaf CO2 accumulation rate (A)”, which makes the specified parameters as the same what it is not true. Results, besides difficult language, are interesting, with good and clear figures, but tables should be completed with the LSD values. In Discussion chapter, the obtained results are cited as they were reported in the Results chapter with numbers of tables and figures. Instead of that they should be generated, clearly explained and interpreted based on the adequate literature. Conclusions are not separated into additional chapter. One questionable thing relates to plant density, which was not tested in the experiment.
The manuscript can be accepted for publication, however the text requires a deep revision of the grammar in order to make it easier to follow and methodological corrections.

Author Response
Dear the reviewer,
Thank you for your many valuable suggestions. I was able to notice many errors and shortcomings. I have made corrections based on your suggestions, and I would appreciate your guidance again.
Sincerely,
Point 1: The manuscript is written with the English language including many grammar mistakes, which make the text difficult to follow.
Response 1: Corrections were made to the parts that were pointed out. In addition, we are currently reviewing the English version.
Point 2: Material and Methods chapter needs to be corrected. I would suggest changing the name of the “no application treatment” into just “control treatment” or “control” and to precise N rate in the other treatments and explanation it’s the base (how it was calculated).
Response 2: No fertilization treatment was changed to control, and a description of N rate was added.
Point 3: PAR parameter should not be recognized as “morphological” one and moved to other sup chapter.
Response 3: The subtitle of this chapter was changed to Measurement of canopy structure because the measurement of PAR is part of the stratified clip method.
Point 4: Among many mistakes one is necessary to explain: the photosynthesis parameter measured with LI-6400 Photosynthesis System is named by the author as just “photosysnthesis (A)” or “Maximum net leaf CO2 accumulation rate (A)”, which makes the specified parameters as the same what it is not true.
Response 4: The words used have been standardized to "photosynthetic rate" as used in the Li-6400 manual. The abbreviation "A" is used. The photosynthetic rate at light saturation was defined as Amax.
Point 5: Results, besides difficult language, are interesting, with good and clear figures, but tables should be completed with the LSD values.
Response 5: In materials and methods, it is stated that the Tukey method was used for multiple comparison, so adding LSD in table, which is used in Fisher's LSD method, causes inconsistency between the text and the table. Therefore, we did not change it.
Point 6: In Discussion chapter, the obtained results are cited as they were reported in the Results chapter with numbers of tables and figures. Instead of that they should be generated, clearly explained and interpreted based on the adequate literature. Conclusions are not separated into additional chapter.
Response 6: A new section "conclusion" was added. In the Discussion section, the comparisons between the results obtained in this study and those in the citations were added.
Point 7: One questionable thing relates to plant density, which was not tested in the experiment.

Response 7: The relationship between this study and the cited references that are the basis for adding the description of planting density to "conclusion" was described.
Reviewer 2 Report
Dear Authors,
The subject of the study is interesting and topical, with high scientific and practical importance.
The introduction is presented correctly, in accordance with the subject. Numerous scientific articles, in concordance to the topic of the study, were consulted.
Methodology of the study was clearly presented, and appropriate to the proposed objectives.
The obtained results are important and have been analyzed and interpreted correctly, in accordance with the current methodology.
The discussions are appropriate, in the context of the results, and was conducted compared to other studies in the field.
The scientific literature, to which the reporting was made, is recent and representative in the field.
Some suggestions and corrections were made in the article.
The following aspects are brought to the attention of the authors.
1.
“Manihot esculenta Crantz” instead of “Manihot esculenta Crantz”
It is recommended that the species name be written in italics
2.
Abstract
The Abstract is too narrative. It can be improved by presenting a summary of significant results, according to Instructions for Authors, and Microsoft Word template, Agronomy journal.
3.
Citing bibliographic sources in the text of the article
According to Instructions for Authors, and Microsoft Word template, Agronomy journal:
"References must be numbered in order of appearance in the text (including citations in tables and legends) and listed individually at the end of the manuscript. We recommend preparing the references with a bibliography software package, such as EndNote, ReferenceManager or Zotero to avoid typing mistakes and duplicated references. Include the digital object identifier (DOI) for all references where available."
Eg
Page 1, row 34,
"[1]" instead of "[14]"
The entire article needs to be revised for the correct citation of bibliographic sources.
Also, simultaneously with the correction of the citation of the bibliographic sources in the article, the References chapter must be corrected, in order to be concordance.
4.
Different way to write values and units of measurement
eg
Page 11, row 265
“240cm” - without space
Page 11, row 266
“240 cm” - with space
There are other such situations throughout the article, at other values and units of measurement.
Revision and correction are required for a unitary presentation, in accordance with Instructions for Authors, and Microsoft Word template, Agronomy journal.
5.
Various other corrections and suggestions
“20 °C”
“100 kg ha-1”
“30 m2”
“CO2”
“m2 s-1”
“mol-1”
6.
References
According to Instructions for Authors, and Microsoft Word template, Agronomy journal:
"References must be numbered in order of appearance in the text (including citations in tables and legends) and listed individually at the end of the manuscript. We recommend preparing the references with a bibliography software package, such as EndNote, ReferenceManager or Zotero to avoid typing mistakes and duplicated references. Include the digital object identifier (DOI) for all references where available."
The entire References chapter needs to be revised.
The bibliographic sources will be presented in the order of citation in the text.
The bibliographic source "14" (current order) will be the bibliographic source "1", as quoted in the text of the article.
According to Instructions for Authors, and Microsoft Word template, Agronomy journal,
- Author 1, A.B.; Author 2, C.D. Title of the article. Abbreviated Journal Name Year, Volume, page range.
Sample:
a)
“Adelekan, B.A. Investigation of ethanol productivity of cassava crop as a sustainable source of biofuel in tropical countries. Afr. J. Biotechnol. 2010, 30, 5643-5650.“
Instead of
“Adelekan, B. A. Investigation of ethanol productivity of cassava crop as a sustainable source of biofuel in tropical countries. African Journal of Biotechnology, 2010, 30, 5643-5650.”
b)
“Bakayoko, S.; Tschannen, A.; Nindjin, C.; Girardin, O. Impact of water stress on fresh tuber yield and dry matter content of cassava (Manihot esculenta Crantz) in Côte d'Ivoire. Afr. J. Agric. Res. 2009, 4, 21-27. “
Instead of
“Bakayoko S.; Tschannen A.; Nindjin C.; Girardin O. Impact of water stress on fresh tuber yield and dry matter content of cassava (Manihot esculenta Crantz) in Côte d'Ivoire. African J of Agric Res, 2009, 4, 21-27. “
c)
“Cadavid, L.F.; El-Sharkawy, M.A.; Acosta, A.; Sánchez, T. Long-term effects of mulch, fertilization and tillage on cassava grown in sandy soils in northern Colombia. Field Crops Res. 1998, 57, 45-56. https://doi.org/10.1016/S0378-4290(97)00114-7”
Instead of
“Cadavid, L.F., El-Sharkawy, M.A., Acosta, A. & Sánchez, T. 1998. Long-term effects of mulch, fertilization and tillage on cassava grown in sandy soils in northern Colombia. Field Crops Res., 57, pp. 45-56.”
It is recommended to pay attention to:
The comma after the name of each author (eg “Bakayoko, S.; instead of “Bakayoko S.;”)
Correct abbreviation of journals name (eg. “Afr. J. Agric. Res.” instead of “African J of Agric Res”)
The year in bold style (eg. “2010” instead of “2010”)
Doi number where possible (eg. “https://doi.org/10.1016/S0378-4290(97)00114-7”)
More suggestions and corrections were made in the article.

Author Response
Dear the reviewer,
Thank you for your many valuable suggestions. I was able to notice many errors and shortcomings.
I have made corrections based on your suggestions, and I would appreciate your guidance again.
Sincerely,
Point 1: “Manihot esculenta Crantz” instead of “Manihot esculenta Crantz”
Response 1: They were corrected.
Point 2: Abstract
The Abstract is too narrative. It can be improved by presenting a summary of significant results, according to Instructions for Authors, and Microsoft Word template, Agronomy journal.
Response 2: The abstract was modified according to the template.
Point 3: Citing bibliographic sources in the text of the article
Response 3: We reviewed the cited references and entered the digital object identifier (DOI). We are currently using Mendeley to organize citations. We will reflect them in the manuscript soon.
Point 4:
Different way to write values and units of measurement
Response 4: The notation of values and measurement units was unified.
Point 5: Various other corrections and suggestions
Response 5: Notation such as superscripts and italics was revised.
Point 6: References
Response 6: We reviewed the cited references and entered the digital object identifier (DOI). We are currently using Mendeley to organize citations. We will reflect them in the manuscript soon.
Point 7:
Response 7: The list of references was modified according to the template.